# The Gouty Kidney: A Reappraisal

**Thomas Bardin [1,2,*], Emmanuel Letavernier [3,4] and Jean-Michel Correas [5]**

1   INSERM U1132 Unit, Hôpital Lariboisière, Université de Paris-Cité, 75010 Paris, France
2   French-Vietnamese Research Centre on Gout and Chronic Diseases, Vien Gut Medical Center, Ho Chi Minh City 70000, Vietnam
3   Service des Explorations Fonctionnelles Multidisciplinaires, Assistance Publique-Hôpitaux de Paris, Hôpital Tenon, 75020 Paris, France
4   Unité Mixte de Recherche (UMR) S 1155, Institut National de la Santé et de la Recherche Médicale, Hôpital Tenon, Sorbonne Université, 75020 Paris, France
5   Radiology Department, Hôpital Necker-Enfants Malades, Université Paris-Cité, 75015 Paris, France
*   Correspondence: thomas.bardin.6@gmail.com

**Abstract:** This review re-examines the role of crystal deposition in the kidney in view of recent clinical and experimental findings. The involvement of the renal system in gout seems frequent. Indeed, recent studies showed that approximately 25% of patients with gout experience renal failure, defined by estimated glomerular filtration rate <60 mL/min/1.73 m$^2$. The pathophysiology is complex and involves several factors, their respective roles being difficult to dissect. The role of crystal deposition in the kidney was the first suspected, and the concept of gouty microcrystalline nephropathy, also called gouty nephropathy, has been popular, supported by early autopsy studies demonstrating uric acid and urate crystal deposition in the renal medulla of patients with gout, together with features of tubulointerstitial nephritis. Crystal deposition was first considered an important source of renal involvement in gout. After the introduction of urate-lowering drugs and the performance of kidney biopsies, which mainly involved the renal cortex and did not reveal much crystal deposition but rather vascular changes, this concept has been criticized and even dismissed. Thereafter, kidney involvement in gout was considered mainly vascular, related to hypertension and associated comorbidities and later to hyperuricemia. The toxic effects of non-steroidal anti-inflammatory drugs is also an important factor. Modern imaging, especially renal ultrasonography, allows for atraumatic exploration of the kidney and has revealed hyperechogenicity of the renal medulla, suggesting crystalline deposits, in approximately one-third of patients with tophaceous gout. Experimental models of gouty nephropathy have recently demonstrated the pathogenic role of microcrystal deposition in the collecting ducts and parenchyma of the renal medulla. Taken together, these recent findings lead to the re-examination of the pathogenic role of crystal deposition in the renal medulla and testing the effect of urate-lowering drugs on renal features of gouty patients with evidence of renal crystal deposition.

**Keywords:** gout; uric acid crystals; urate crystals; autopsy; renal ultrasonography; hyperechoic medulla; spectral computed tomography; experimental models of gouty nephropathy

## 1. Introduction

Patients with gout frequently experience chronic renal disease (CKD), which substantially complicates the gout management and worsens the prognosis [1]. A meta-analysis of 9 studies found a 24% (95% confidence interval [CI]: 19 to 28) prevalence of CKD stage $\geq$3 (estimated glomerular filtration rate [eGFR] < 60 mL/min/1.73 m$^2$, the usual clinical definition of renal failure) [2]. In 2009–2010, an estimated 7.58 million people were affected by gout in the United States, and among these, 5.86 million had CKD, defined by eGFR <90 mL/min/1.73 m$^2$ [3]. In this cross-sectional study, gout prevalence increased with CKD stage.

Renal failure is a well-accepted cause of gout because it decreases the output of uric acid in the urine and therefore favors hyperuricemia and gout. However, this relation is bidirectional, and gout is also a causal factor for CKD. In a large English retrospective cohort, gout was associated with increased risk of incident CKD stage ≥3 in the 3 years after gout diagnosis as compared with patients without gout (adjusted hazard ratio: 1.78; 95% CI: 1.70 to 1.85) [4]. Gout has also been associated with an increased risk of advanced chronic kidney disease [5,6]. In a national Taiwanese cohort, gout increased the risk of progression to end stage renal disease (hazard ratio: 1.57; 95% CI: 1.38 to 1.79) [5].

The causes of kidney involvement in gout seem complex, multiple and difficult to dissect. Historically, crystal deposits demonstrated at autopsy in the renal medulla have been considered the central cause. After urate-lowering drugs became available, these deposits were seen less frequently. Renal biopsies, which targeted the renal cortex and avoided the medulla and its rich vascularization, did not reveal crystal deposits but rather showed signs of vascular nephropathy. Of note, most kidney biopsies are exposed to aqueous formalin fixative, which results in monosodium urate (MSU) crystal dissolution and therefore this finding may be underscored [7]. Thus, the role of crystals in the pathogenicity of kidney involvement in gout was largely dismissed. Renal vasculopathy induced by hypertension and other comorbidities such as type 2 diabetes mellitus and lipid disorders were considered the main or exclusive pathogenic mechanism [8–10]. More recently, the detrimental effects of non-steroidal anti-inflammatory drugs (NSAIDs) and excessive soluble uric acid were highlighted [11].

This review re-examines the role of crystal deposition in the kidney, a condition historically called the gouty kidney [12,13], in view of recent clinical and experimental findings.

## 2. Early Works: Autopsy Findings

Crystal deposits in the kidneys of patients with gout have been reported since the 1800s [14,15]. A recent review found 25 reports of crystal deposits in the kidney by autopsy, biopsy or imaging [16]. The most numerous reports dealt with early autopsy studies performed in the 1950s to 1970s, before the extensive use of effective urate-lowering drugs [17–25]. The main crystal deposition site was the renal medulla [18,22,23]. In most reports, interstitial typical MSU crystal deposits (tophi) and intratubular granular uric acid deposits were observed by light microscopy. The hypothesis that uric acid crystal deposits could lead to tubular membrane lysis and liberation of uric acid crystals in the interstitium of the medulla, and then by a nucleation process could secondarily lead to MSU crystal deposits was popular among pathologists and supported by the observation of oxonic acid-treated hyperuricemic rats [13]. However, in 1966, Seegmiller and Frazier reported the results of their study of renal crystals obtained on autopsy of gouty patients; by using X-ray diffraction, the authors identified monosodium urate in the pyramid deposits (or sometimes calcium oxalate). They did not find uric acid crystals, which were in contrast abundant in tubular deposits of non-gouty leukemic patients with tumor lysis syndrome [23]. Therefore, Seegmiller and Frazier proposed the alternative hypothesis that interstitial MSU crystals could form in the medulla with increased supersaturation of urate and sodium following the renal concentration processes toward the papilla [23]. Medullary tophi appeared surrounded by giant cells. Cellular infiltration of the interstitium was observed, thus leading to a diagnosis of "sterile pyonephrosis" in some of these early reports, and coexisted with fibrosis and tubular alterations. These lesions were held responsible for chronic tubulointerstitial nephritis.

These early autopsy studies also often revealed glomerulosclerosis and arteriolosclerosis and suggested that vascular nephropathy due to the frequent association of hypertension could play an important part in the renal decline frequently observed in gouty patients. More recently, the possible role of excessive soluble uric acid in the observed vascular changes has been emphasized and supported by animal studies [11].

Medullary tophi in these autopsy studies were not specific to gout, and this subsequently appeared as an important limitation to the concept of gouty kidney or urate nephropathy.

Medullary tophi were also seen in patients with renal failure but no gout [20,24] and, more rarely, in patients with no gout or renal failure [24]. These findings were recently confirmed by the study of renal biopsies, which predominantly or exclusively involved the renal medulla in one nephrology center. Medullary tophi were seen in 36 of 796 such biopsies, all but 2 from hyperuricemic patients and all but 1 from CKD patients. According to the authors, in impaired kidney function, the uric acid-filtered per-remnant nephrons would be increased in number, thus leading to a higher concentration of uric acid in the collecting duct, favoring crystal formation. Because each collecting duct is connected to >2000 nephrons, their obstruction by crystals could considerably worsen kidney function [25]. Following this hypothesis, crystal deposits could worsen CKD of a gouty and non-gouty origin.

## 3. Imaging Studies

### 3.1. Kidney Ultrasonography (US)

On B-mode US, the renal medulla appears hypoechoic as compared with the surrounding renal cortex in healthy kidneys, the so-called cortico-medullar differentiation (Figure 1). Reverse corticomedullary differentiation (hyperechoic medulla) (Figures 2 and 3) has been reported as a reversible finding in neonates, and is also a well-admitted feature of several specific diseases, including medullary nephrocalcinosis resulting from hyperparathyroidism, sarcoidosis, vitamin D toxicity, medullary sponge kidney, sickle cell anemia, type 1 renal tubular acidosis, Tamm–Horsfall proteinuria, recessive polycystic disease and hemoglobinuria [26–29].

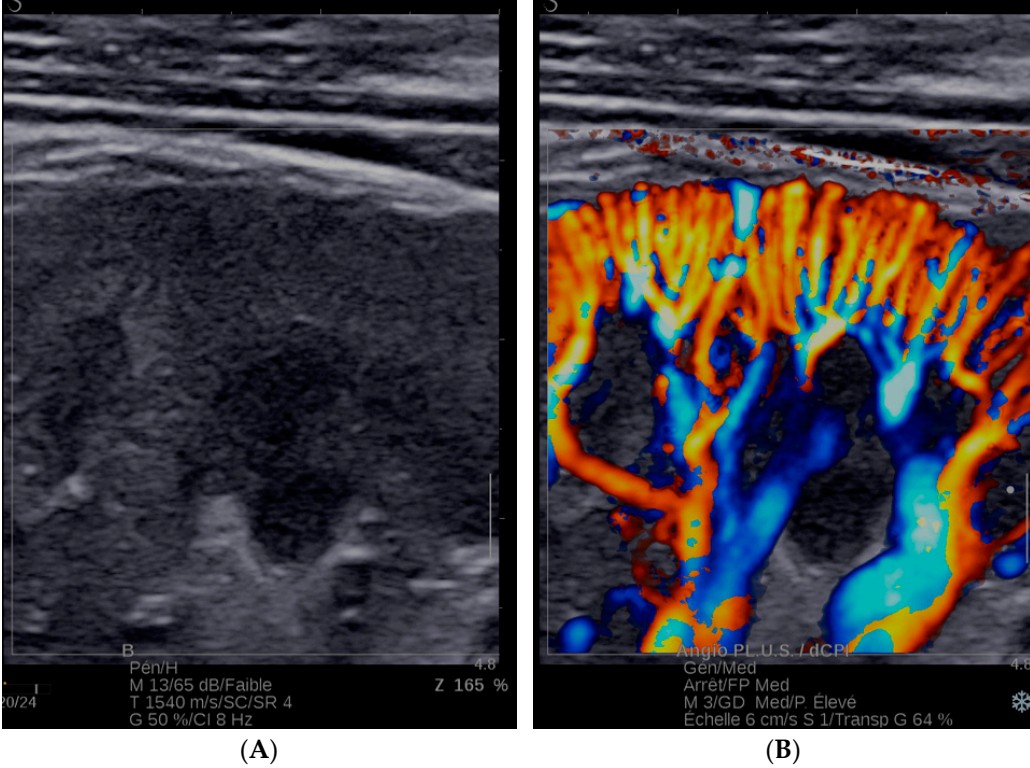

**(A)**                                        **(B)**

**Figure 1.** Kidney ultrasonography (US) of a normal kidney; linear low frequency transducers were here used to improve the visualization of renal cortex and medulla, as well as the detection of intraparenchymal vessels. (**A**) B mode US: the renal medulla is developed in the deep parenchyma as hypoechoic pyramids facing the renal sinus; their echogenicy is lower than the one of the renal cortex. (**B**) The vessels surrounding the medulla (i.e., the interlobar and the arcuate arteries) are detected at directional power Doppler ultrasound. No twinkling artefact is observed.

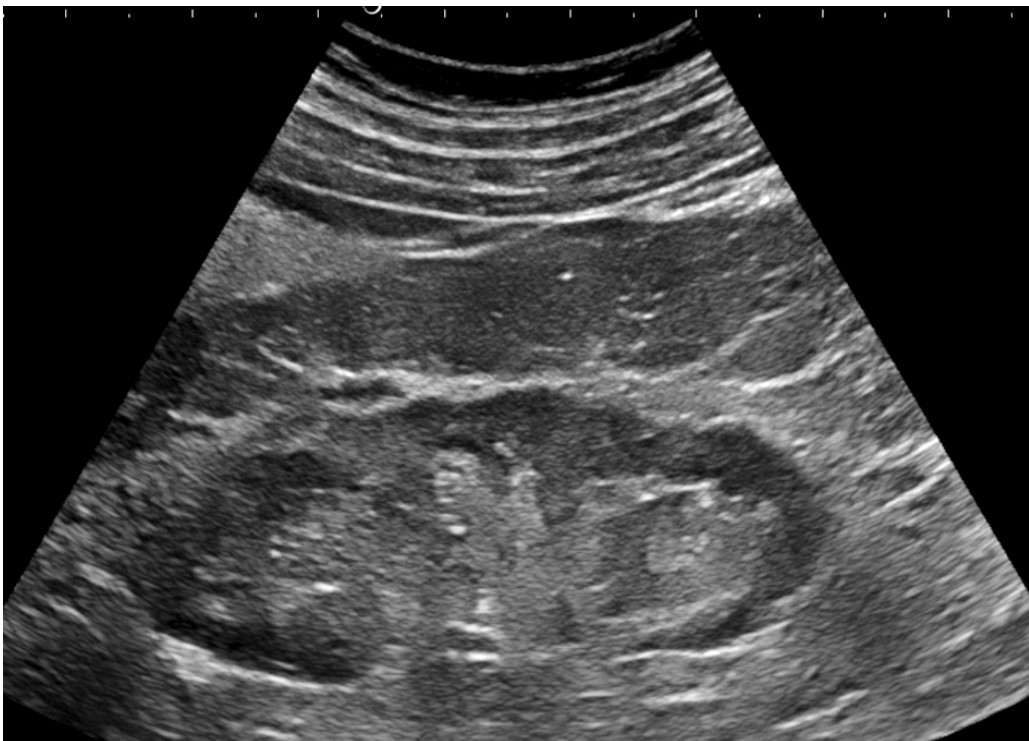

**Figure 2.** Mild gouty nephropathy. B mode US: the medulla is hyperechoic reversing the typical cortico-medullary differentiation.

Several authors reported a hyperechogenic pattern of the renal medulla on B-mode US in Asian patients with gout [30–32]. In a review of 8000 renal US scans consecutively performed in a Japanese hospital over 2 years, hyperechogenicity of the medulla was observed in only 18 patients, 7 of whom had a diagnosis of gout and 1 of Lesch–Nyhan syndrome [30]. The hyperechoic pattern of the medulla has been repeatedly observed in this syndrome and explained by oxypurine deposits in the medulla, including xanthine in allopurinol-treated patients [33,34]. In our cross-sectional study of 502 consecutive Vietnamese patients with untreated gout and a median estimated disease duration of 4 years (interquartile range 0–27), we observed diffuse hyperechoic medulla in 181 patients (36%; 95% CI: 32 to 40) that was not seen in 515 consecutive patients without gout observed during the same time [32]. According to univariate analysis, the hyperechoic pattern was associated with patient age, estimated gout duration, corticosteroids dependency, presence of clinical tophi, double contour thickness, urate arthropathy (all $p < 0.0001$), coronary heart disease ($p = 0.006$), hypertension ($p = 0.0008$), hyperuricemia ($p = 0.002$), urine-stick proteinuria ($p = 0.0006$), leukocyturia ($p = 0.0008$) and decreased eGFR ($p < 0.0001$). From multivariate analysis, the pattern was associated with estimated gout duration (odds ratio (OR), 2.13; 95% CI, 1.55 to 2.96; $p < 0.0001$), clinical tophi (OR, 7.27; 95% CI, 3.68 to 15.19; $p < 0.0001$), urate arthropathy (OR, 3.46; 95% CI, 1.99 to 6.09; $p < 0.0001$), double contour thickness (OR, 1.45; 95% CI, 1.06 to 1.97; $p < 0.02$) and eGFR (OR, 0.30; 95% CI, 0.09 to 0.89; $p < 0.034$). Decreased renal function seemed mild in patients with a hyperechoic medulla with a median eGFR of 70 mL/min/1.73 m$^2$ (95% IR: 60 to 83) as compared with patients with no hyperechoic medulla (81 mL/min/1.73 m$^2$; IR: 71 to 90). To ensure that our findings were not limited to Vietnamese patients, we also performed renal US in a short series of 10 patients with tophaceous untreated, very poorly or recently (<3 months) treated gout seen at the Lariboisieère Hospital in Paris. Four patients exhibited diffuse hyperechogenicity of the renal medulla on B-mode US. On color Doppler US, numerous twinkling artifacts were observed in the hyperechoic medullae of our Vietnamese and French patients. These artifacts have been described as a mosaic color pattern behind a strongly reflective irregular structure [35]. They have been reported primarily in association with crystalline urinary

stones [36,37] and can be reproduced in vitro by synthetic crystalline structures [38], so they can be considered indicators of crystalline deposition.

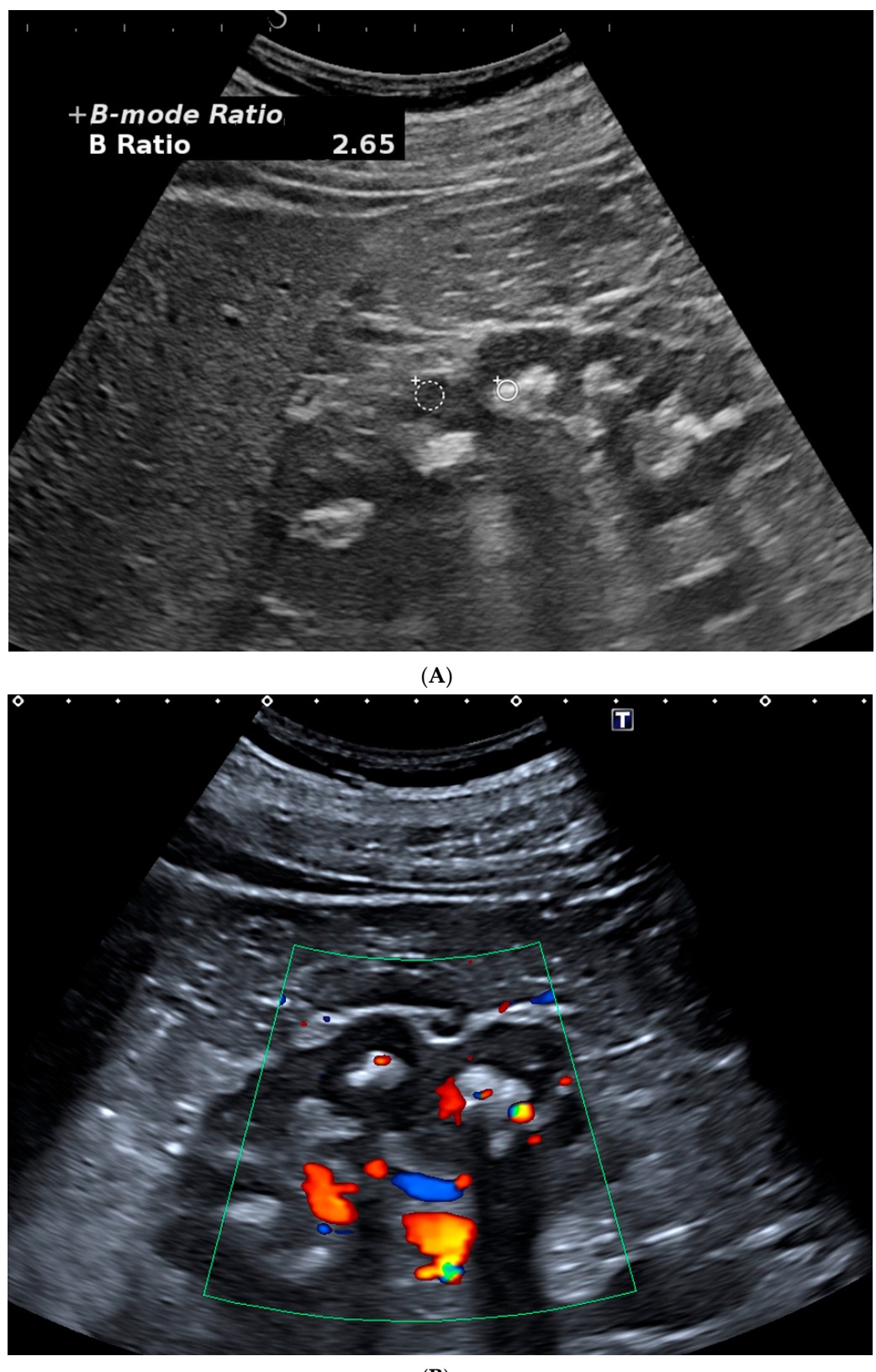

(**A**)

(**B**)

**Figure 3.** Severe gouty nephropathy. (**A**) B mode US: the medulla becomes strongly hyperechoic. (**B**) Power Doppler US: twinkling artifact can be detected.

*3.2. Spectral CT*

We have also identified medullary uric acid/urate medullary deposits by spectral CT (also called dual energy CT) in two patients with gout and strongly hyperechoic medullae [39]. Spectral CT allows for identifying urate in peripheral joints with good specificity [40] but has some limitations. Even though our finding needs to be expanded by using a carefully designed spectral CT protocol adapted to the characterization of crystalline deposition in the kidney, we believe that the exact superposition of the US and spectral CT images in these two patients strongly argues against artifactual findings and supports the hypothesis that hyperechoic medullae observed in gouty patients reflects urate deposition.

In vivo imaging studies suggest that crystal deposition in the renal medulla frequently occurs in untreated patients with severe tophaceous gout [41] and is associated with renal impairment and features of tubulointerstitial nephritis as observed in the early autopsy studies. In this regard, the recent report of the association of tophi with renal involvement in patients with gout can be taken as further support of crystal deposition in the kidney having a pathogenic role [42].

## 4. Gout and Microcrystalline Nephropathies

A number of authors have classified gouty nephropathy, characterized by the deposition of uric acid/urate crystals in the renal medulla, into the broad spectrum of microcrystalline nephropathies. These include a variety of well-accepted conditions characterized by crystal deposition in the renal parenchyma. In addition to uric acid-derived crystals, deposition of calcium phosphate, calcium oxalate, monoclonal immunoglobulins, several drugs (antiproteases, methotrexate, acyclovir, sulfadiazine, etc.) and other crystal types can cause chronic CKD [43,44]. These crystals can also be source of acute renal failure, as seen after the massive precipitation of immunoglobulin light chains in the renal tubules [45]. Similarly, acute uric acid nephropathy, although not a feature of gout, can follow the tubular precipitation of uric acid crystals during tumor lysis syndrome, produced by a massive production of uric acid during chemotherapy treatment of malignant tumors or leukemias [46]. In addition, a number of crystals implicated in the genesis of microcrystalline nephropathies, including calcium phosphate and oxalate crystals, can produce renal lithiasis, in a similar fashion to gout. Indeed, uric acid nephrolithiasis is a frequent complication of gout [47], favored by low urine pH and in most cases increased uric acid renal excretion; a meta-analysis of 10 studies estimated the lifetime prevalence of self-reported nephrolithiasis at 14% (95% CI: 12 to 17) [2]. Metabolic syndrome causes decreased fractional excretion of uric acid, increased uric acid serum levels and risk of gout. Patients with metabolic syndrome and diabetes also have decreased ammonia genesis in the proximal tubule. This defect results in low urine pH, frequently < 5.3, inducing uric acid precipitation and uric acid stone formation, even without increased urine uric acid excretion [48]. The underlying metabolic syndrome or diabetes also explains why gouty patients are at risk of uric acid stones.

## 5. Experimental Models of Gouty Nephropathy

Several murine models of gouty nephropathy have been developed [49]. Rodents possess the urate oxidase (Uox) enzyme, which converts uric acid into the more soluble allantoine, thus resulting in very low uricemia. The Uox enzyme is mainly located in hepatocytes, where uric acid enters by using glucose transporter 9 (GLUT9), which, in rodents, is expressed on hepatocyte membranes [50]. Several ways of inhibiting Uox in rodents have been used to produce animal models of gouty nephropathy: inhibiting urate oxidase by oxonic acid, gene deletion of urate oxidase or inhibiting the GLUT9-mediated transport of uric acid into hepatocytes. These experimental models schematically mainly produce acute nephropathy (uric acid-blockade acute nephropathy), which can be followed by a second chronic phase, the pathological features of which are close to the human chronic gouty nephropathy.

### 5.1. Inhibition of XO by Oxonate

The first rat model that used oxonate to inhibit urate oxidase and uric acid supplements to further increase uricemia and uricuria [51,52] illustrates this two-step evolution. Male Wistar rats fed oxonic acid (0.4 mg/day) and uric acid (0.6 g/day) became hyperuricemic in 3 to 4 weeks and exhibited extremely high uricosuria and acute renal failure. This was pathologically characterized at 1 month by amorphous uric acid deposits in the collecting ducts predominantly at the papillae, tubular injury and an exudative response consisting of neutrophilic granulocytes, as well as early interstitial MSU crystal tophus formation [52]. This early uric acid-blockade acute nephropathy was followed by recovery of renal function. However, after 36 and 52 weeks of hyperuricemia, the kidneys showed a predominantly interstitial mononuclear cell infiltrate around regenerated tubules, interstitial fibrosis, infrequent renal tophi and renal stones, with no evidence of glomerular or vascular abnormalities and no gross evidence of renal failure. A mouse model of intragastric administration of oxonic acid and adenosine (a precursor to uric acid) was recently used to reproduce this model of acute gouty nephropathy [53].

### 5.2. Urate Oxidase-Knockout Mice

Several methods of Uox-knockout have been used in mice and led to the deposition of acid uric/urate crystals in the kidney. By inserting a neomycin cassette into exon 3 of Uox in embryonic stem cells, Wu et al. were the first to produce Uox-knockout mice. Uricuria and uricemia were very high, the latter ranging from 200 and 660 µmol/L, acute and severe microcrystalline nephropathy developed early, and the mortality rate was high (65% at 4 weeks) [54]. Using the transcription activator-like effector nuclease (TALEN) technology, Lu et al. generated a Uox-knockout mouse with less severe hyperuricemia (420 to 520 µmol/L), in the same range as the serum uric acid level of hyperuricemic humans, with better longevity [55]. At 6 weeks, Uox-knockout mice exhibited nephropathy with elevated creatinine and urea levels, dilated Bowman's spaces and tubules, collapsed and necrotic nephrons, focal tubulointerstitial fibrosis, severe non-specific chronic corticomedullar inflammation with lymphocyte and macrophage infiltration, increased expression of inflammatory cytokines and deposition of urate crystals in the kidney. Urate-deficient rats have more recently been produced by using the CRISPR/Cas9 technique [56]. These rats had higher uricemia than wild-type animals but levels were still within the normal range for humans and they exhibited only minimal kidney lesions, with sporadic interstitial fibrosis, inflammatory cell infiltration and no crystal deposition.

### 5.3. SLC2A9-Knockout Mice

Methods to inhibit the access of uric acid to the liver Uox focused on the Glut9 transporter, encoded by the *SLC2A9* gene. Glut9 is expressed in the mouse liver, where it transports uric acid inside hepatocytes, and in mice and humans in the kidney, where it reabsorbs uric acid from the urine. Mice with systemic *SLC2A9* knockout exhibited moderate hyperuricemia, massive hyperuricosuria and early-onset nephropathy, starting at 2 weeks. The nephropathy was characterized by mild renal insufficiency, obstructive crystal deposits of the medullary tubules, microcysts in the tubules and hydronephrosis, tubulointerstitial inflammation and progressive inflammatory fibrosis of the cortex [50]. Selective depletion of liver Glut9 was produced by tamoxifen injections (LG9KO mice) and resulted in lower uricuria (about 120 µmol/L) than systemic knockout, explained by the conservation of renal Glut9, which reabsorbed uric acid from the urine. This latter mouse model exhibited no crystallization of uric acid in the kidney and no urate nephropathy. To further increase the uricemia, LG9KO mice were gavaged with inosine (4 g/day), a uric acid precursor, for 3 days and fed a chow or high-fat diet [57]. In chow-fed mice, uricemia increased transiently at about 300 µmol/L, 3 days after the start of inosine and returned to baseline levels 24 h after stopping the inosine. Urine pH remained stable. No renal failure developed. Renal morphology was normal except for minimal tubular crystal deposition. In mice fed a high-fat diet, the uricemia peak was greater

(about 500 µmol/L), urine pH decreased to 5.6 and plasma creatinine level increased five-fold, indicating acute renal failure. Pathological examination of the kidneys 3 days after inosine supplements demonstrated obstructive ammonium urate and uric acid crystal deposition within medullary tubules, tubule dilation, macrophage infiltration and elevated inflammatory cytokine mRNA levels. Six weeks after inosine gavage, plasma urate and creatinine levels had normalized, but mice exhibited renal inflammation, fibrosis, tubular atrophy and renal shrinking, despite the disappearance of urate and uric acid crystals. This model clearly suggests that transient hyperuricosuria in acidic urine provokes transient renal failure by inducing intratubular crystal formation, tubular obstruction and renal inflammation. It also shows that renal remodeling continues to progress despite the disappearance of crystals. Another group of investigators studied the effect of more lasting hyperuricemia in chow-fed mice and mice under an acidogenic diet by prolonging the inosine feeding of mice until sacrifice at day 42, in LG9KO and control mice [58]. They demonstrated the role of crystal formation in the deleterious renal effects of hyperuricemia. Hyperuricemic chow-fed LGKO and control mice, with baseline normal function or with aristolochic acid I-induced nephropathy, exhibited neither acute nor chronic loss of renal function, and renal failure appeared only in mice with uric acid crystalluria favored by urine acidification resulting from the diet. Acute renal failure that developed early was confirmed associated with uric acid crystal deposition in the inner medulla and the induction of granulomatous nephropathy. Follow-up of the crystalluric mice showed a further decline of renal function. Mice exhibited chronic uric acid crystalline nephropathy, similar to human chronic urate nephropathy, characterized by crystal deposition in the medulla and granulomatous interstitial nephritis. Of note, granuloma did not include tubular epithelial cells and appeared to form after the onset of interstitial fibrosis. Various immune cells, namely proinflammatory M-1 macrophages, neutrophils and dendritic cells, were characterized in the observed chronic granulomatous nephritis. Tofacitinib inhibition of Janus kinase/signal transducer and activator of transcription did not affect crystal granuloma formation or CKD progression in this chronic uric acid crystal nephropathy. In contrast, adenosine reduced granuloma formation and CKD progression, which suggests an important role for M-1 macrophages in this animal model [58].

When LG9KO mice were supplemented with incremental amounts of inosine in a normal chow diet, they exhibited a progressive increase in uricemia up to 300 µmol/L. However, they showed no hypertension or cardiovascular abnormalities, despite the appearance of mild renal dysfunction without acidification of urine and crystalline deposits, thus challenging a direct role for soluble uric acid in the control of blood pressure [59]. This observation agreed with experimental results showing that soluble uric acid had an anti-inflammatory effect and promoted the idea that the deleterious effects of uric acid observed in vitro were due to the presence of small amounts of crystals owing to the mode of obtaining uric acid solutions [60]. Results of Mendelian randomization studies and clinical trials also feed the present debate on the role of soluble uric acid in the genesis of hypertension and renal vasculopathy [61,62].

*5.4. Mechanism of Experimental Gouty Nephropathies*

The mechanisms of the experimental uric acid nephropathies can be summarized from the above-mentioned experimental models and what is known about crystal nephropathies in general, as reviewed in [44]. Supersaturation of uric acid in urine, favored by urine acidification, seems to be the first step in uric acid crystal deposition in the medulla distal tubules and collecting ducts. Lumen obstruction is thought to be the primary cause of early acute renal failure. In addition, crystals are directly or indirectly cytotoxic to tubular cells. Tubular cell necrosis releases alarmins, proteases, nucleic acids and damage-associated molecular patterns that can activate Toll-like receptors and trigger the activation of the nucleotide oligomerization domain-like receptor protein 3 (NLRP3) inflammasome of renal dendritic cells. The NLRP3 inflammasome is also directly activated by the crystals. This leads to the release of active interleukin 1 (IL-1) and IL-18, which

drive acute inflammation, thus resulting in interstitium infiltration of polymorphonuclear neutrophils and mononuclear cells that produce a variety of inflammatory mediators. Pro-inflammatory M-1 macrophages predominate and produce IL-6 and tumor necrosis factor α, the latter worsening tubular cell death. Damaged tubular cells, regularly observed in animal models, release a variety of profibrotic mediators, including transforming growth factor β, integrin-linked kinase, β-catenin, Notch1 and hypoxia-inducible factor 1 [63], which likely play a role in the early onset of interstitial fibrosis observed by Sellmayr et al. [58]. In addition, activation of NLRP3 promotes transforming growth factor β receptor signaling by a pathway independent of IL-1 activation and contributes to fibrosis [44].

Experimental chronic uric acid microcrystalline nephropathy follows the acute phase if hyperuricemia is maintained in living animals. This chronic phase is characterized by the deposition of uric acid crystals and granuloma formation in the interstitium. Crystal deposits are then free of tubular cell markers [58], and in an early experimental model using oxonate, crystals seemed to migrate from the tubule to the interstitium [13]. An experimental model of 2,8-dihydroxyadenine (2,8 DHA) nephropathy, a rare disease caused by hereditary deficiency of adenine phosphoribosyltransferase leading to tubular deposition of 2,8 DHA crystals, has allowed for better understanding how pathogenic crystals can migrate from the tubule lumen to the interstitium [64]. In this model, sequential pathological examination demonstrated that medium-sized 2,8 DHA crystals induced a reparative process that the authors termed extratubulation. Tubular cells, in coordination with macrophages, overgrew and translocated crystals into the interstitium, thus restoring the tubular lumen; interstitial crystals were then degraded by granulomatous inflammation. This phenomenon explains the observed improvement in renal function and the presence of pathogenic crystals in the interstitium. Interstitial crystal deposits drive the formation of granulomas, which include giant cells formed by the fusion of several macrophages, which could be triggered by their inability to eliminate the crystals, and a variety of inflammatory cells, including macrophages predominantly of the M-1 proinflammatory type. M-1, like macrophages, seem to play an important role in the progression of experimental, chronic granulomatous uric acid nephropathy, as shown by the positive result of suppressing their activation by adenosine, which resulted in decreased fibrosis and granuloma formation and increased GFR [58]. Of note, targeting the renal inflammation by steroids or an IL-1 receptor antagonist increased renal function recovery in the 2,8 DHA model [65].

### 5.5. Comparison to Human Gouty Nephropathy

These animal models support a critical role for crystal formation and deposition in the renal toxicity of hyperuricemia and have greatly enlightened our understanding of the effects of crystal deposition in the renal medulla. However, they differ from human chronic gouty nephropathies on at least two points. First, experimental uric acid nephropathies start with the acute deposition of uric acid in distal tubules and collecting ducts, which is not a usual feature of human gout, even though it has been rarely observed in heavily overproducing young boys with Lesch–Nyhan syndrome [66]. This early phase of animal models seems rather a good model of the tumor lysis syndrome. Second, the nature of uric acid-derived crystals was carefully determined in two recent reports [57,58]. Crystals were mainly identified as uric acid, and no monosodium urate was reported; however, the latter was repeatedly observed by light microscopy in the medullary interstitium in early autopsy studies and precisely identified by X-ray diffraction [23]. In our renal US study, a hyperechoic medulla suggested the presence of crystal deposits, associated with peripheral tophi and hyperuricemia but not acidic urinary pH, elevated uric acid-to-creatinine ratio, or fractional clearance of urate, which was low in most patients [32]. This observation may suggest that medullary crystal deposits indeed mainly consisted of monosodium urate deposits even though this hypothesis needs to be validated by pathologic studies. In addition, if uric acid crystal precipitation in the renal tubule lumen would be the primary event leading to urate nephropathy, this condition should affect mostly patients with recurrent uric acid stones. To our knowledge, such an association has not been shown.

## 6. Conclusions

Recent findings led to challenging the role of hyperuricemia without crystal formation in the renovascular disease of patients with gout and to re-evaluate the pathogenic role of microcrystal deposits in the renal medulla of gouty patients. Ultrasonography has demonstrated the frequent presence of a hyperechoic medulla, suggesting crystal deposition, in severe tophaceous gout. This microcrystalline nephropathy is associated with renal dysfunction (which appears most often as mild) and features of tubulointerstitial nephritis. Although other factors including NSAID toxicity and renovascular disease play an important role in the decline of renal function in gouty patients, early treatment with urate-lowering drugs is indicated to prevent renal crystal deposition. The hypothesis that this particular microcrystalline type of kidney involvement can be improved by urate-lowering enabling crystal dissolution deserves to be tested in gouty patients with evidence of crystal deposition in the renal medulla.

**Author Contributions:** Conceptualization, T.B.; writing—original draft: T.B.; writing—review and editing: E.L. and J.-M.C.; Figures: J.-M.C. All authors have read and agreed to the published version of the manuscript.

**Funding:** This research received no external funding.

**Institutional Review Board Statement:** Not applicable.

**Informed Consent Statement:** Not applicable.

**Data Availability Statement:** Not applicable.

**Conflicts of Interest:** The authors declare no conflict of interest.

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
