# Peer review of "The Gouty Kidney: A Reappraisal"

_2813-4583, doi:10.3390/gucdd1010004_

Round 1
Reviewer 1 Report
Congratulations for this well-written paper with an interesting and exhaustive review about pathological mecanisms in the relationship between gout and kidney, including data from image studies, animal models, human autopsies...
consider adding reference(s) supporting lines 52-54: PMID 7699635
consider adding this study/reference in line 46: PMID 31462487
Author Response
We sincerely thank Reviewer 1 for his supporting comments. The two references have been added in the revised manuscript.
Reviewer 2 Report
This is an interesting review article that evaluates the nature of renal damage in gouty patients, both by reviewing the pathology, imaging changes, and the animal models that shed light on our current understanding of how gout induces renal damage.
Intro:
It appears there was an error: A meta-analysis of 9 studies found a 24% (95% confidence interval [CI]: 19 to 24) prevalence of CKD stage > 3 [in patients with gout] (estimated glomerular filtration rate [eGFR] < 60 ml/min/1.73 m2, the usual clinical definition of renal failure) (2). The CI was 19-28.
Imaging studies:
Ultrasound:
Set up: Unusual way to open up the paragraph on renal ultrasound “Several authors reported a hyperechogenic pattern of the renal medulla on B-mode US in Asian patients with gout (26-28).” Move this down to line 134 where you specifically discuss Vietnamese patients. With your discussion in this paragraph of “twinkling structures” that could be uric acid deposition, was there any biopsy specimens to confirm that? Or has there been a study evaluating ultrasound results with biopsy/pathology specimens?
Figures 1-3 need legends.
For the Spectral CT, you only have two patients you discuss that you have evaluated. Is there any data in literature about using Spectral CT for gouty changes in kidneys to further support your 2 published cases?
Interesting discussion on the use of animal models and what we can learn from them. It is particularly interesting how inflammatory the responses were, with active cytokine production and inflammasome activation. Do we have any data about such inflammatory responses in our human biopsies? One of the limitations you discussed with the animal models is how acute the injury is, and how that differs from a more chronic indolent path that we see in humans with cumulative years of damage from hyperuricemia and concomitant comorbidities.
Author Response
This is an interesting review article that evaluates the nature of renal damage in gouty patients, both by reviewing the pathology, imaging changes, and the animal models that shed light on our current understanding of how gout induces renal damage.
Intro:
It appears there was an error: A meta-analysis of 9 studies found a 24% (95% confidence interval [CI]: 19 to 24) prevalence of CKD stage > 3 [in patients with gout] (estimated glomerular filtration rate [eGFR] < 60 ml/min/1.73 m2, the usual clinical definition of renal failure) (2). The CI was 19-28.
Answer:
Thank-you very much for pointing out this silly mistake, which has been corrected in the revised manuscript
Imaging studies:
Ultrasound:
Set up: Unusual way to open up the paragraph on renal ultrasound “Several authors reported a hyperechogenic pattern of the renal medulla on B-mode US in Asian patients with gout (26-28).” Move this down to line 134 where you specifically discuss Vietnamese patients. With your discussion in this paragraph of “twinkling structures” that could be uric acid deposition, was there any biopsy specimens to confirm that? Or has there been a study evaluating ultrasound results with biopsy/pathology specimens?
Answers:
- The first sentence of the paragraph has been moved down.
- For ethical reasons, we did not perform renal biopsies in our study, as biopsy of the renal medulla is dangerous , especially in Vietnam. We are not aware of any study that evaluated US results with pathology specimen. However, the fact that twinkling artefacts are seen at the site of crystalline stones, and the in vitro reproduction of these artefacts by crystalline compounds are sound arguments to believe they reveal crystalline deposits as explained in our manuscript.
Figures 1-3 need legends.
Answer:
Our submitted manuscript contained figure legends, which do not appear in the pdf produced by the publisher, maybe because we did some mistake in the submission process… We apologize for that.
For the Spectral CT, you only have two patients you discuss that you have evaluated. Is there any data in literature about using Spectral CT for gouty changes in kidneys to further support your 2 published cases?
Answer:
Indeed, we have been so far able to perform only two spectral CTs in our patients with hyperechoic medulla, because of the covid epidemics and technical problems. We are not aware of other published studies on this topic. As stated in our manuscript, we need more. Prof Correas is presently finalizing a technical protocol for Spectral CT of the kidney, (his unit should shortly acquire a new CT), and we hope to bring more data in the near future.
Interesting discussion on the use of animal models and what we can learn from them. It is particularly interesting how inflammatory the responses were, with active cytokine production and inflammasome activation. Do we have any data about such inflammatory responses in our human biopsies? One of the limitations you discussed with the animal models is how acute the injury is, and how that differs from a more chronic indolent path that we see in humans with cumulative years of damage from hyperuricemia and concomitant comorbidities.
Answer:
There is presently a paucity of data on human specimen and, to the best of our knowledge, no characterization of inflammation with modern techniques on human biopsies of gouty kidney. Most of what was observed in the human gouty kidney is from old autopsy studies. The dogma that this entity did not exist then prevailed, and people lost interest in the disease or a while.